# Development of an Electromyography Signal Acquisition Prototype and Statistical Validation Against a Commercial Device

**DOI:** 10.3390/s24216787

**Published:** 2024-10-22

**Authors:** Erick Guzmán-Quezada, Santiago Lomeli-Garcia, Jorge Velazco-Garcia, Maby Jonguitud-Ceballos, Adriana Vega-Martinez, Juan Ojeda-Galvan, Francisco J. Alvarado-Rodríguez, Fernanda Reyes-Jiménez

**Affiliations:** 1Department of Electromechanics, Universidad Autónoma de Guadalajara, Guadalajara 45129, Mexico; francisco.alvarado@edu.uag.mx; 2Department of Biomedical Electronic Engineering, Universidad Autónoma de Guadalajara, Guadalajara 45129, Mexico; santiago.lomeli@edu.uag.mx (S.L.-G.); jorge.velazco@edu.uag.mx (J.V.-G.); maby.jonguitud@edu.uag.mx (M.J.-C.); adriana.vega@edu.uag.mx (A.V.-M.); juan.ojeda@edu.uag.mx (J.O.-G.); 3Department of Translational Bioengineering, Centro Universitario de Ciencias Exactas e Ingenierías, Universidad de Guadalajara, Guadalajara 44430, Mexico; maria.reyes2825@alumnos.udg.mx

**Keywords:** electromyography, low-cost EMG, device validation

## Abstract

Electromyography (EMG) stands out as an accessible and inexpensive method for identifying muscle contractions on the surface and within deeper muscle tissues. Using specialized electronic circuits for amplification and filtering can help develop simple but effective systems for detecting and analyzing these signals. However, EMG devices developed by research teams frequently lack rigorous methodologies for validating the quality of the signals they record compared to those obtained by commercial systems that have undergone extensive testing and regulatory approval for market release. This underscores the critical need for standardized validation techniques to reliably assess the performance of experimental devices relative to established commercial equipment. Hence, this study introduces a methodology for the development and statistical validation of a laboratory EMG circuit compared with a professional device available on the market. The experiment simultaneously recorded the muscle electrical activity of 18 volunteers using two biosignal acquisition devices—a prototype EMG and a commercial system—both applied in parallel at the same recording site. Volunteers performed a series of finger and wrist extension movements to elicit myoelectric activity in these forearm muscles. To achieve this, it was necessary to develop not only the EMG signal conditioning board, but also two additional interface boards: one for enabling parallel recording on both devices and another for synchronizing the devices with the task programmatically controlled in Python that the volunteers were required to perform. The EMG signals generated during these tasks were recorded simultaneously by both devices. Subsequently, 22 feature indices commonly used for classifying muscular activity patterns were calculated from two-second temporal windows of the recordings to extract detailed temporal and spatial characteristics. Finally, the Mean Absolute Percentage Error (MAPE) was computed to compare the indices from the prototype with those from the commercial device, using this method as a validation system to assess the quality of the signals recorded by the prototype relative to the commercial equipment. A concordance of 87.6% was observed between the feature indices calculated from the recordings of both devices, suggesting high effectiveness and reliability of the EMG signals recorded by the prototype compared to the commercial device. These results validate the efficacy of our EMG prototype device and provide a solid foundation for the future evaluation of similar devices, ensuring their reliability, accuracy, and suitability for research or clinical applications.

## 1. Introduction

Among the various types of biosignals that can be analyzed non-invasively, electromyography (EMG) signals are considered one of the most notable due to their extensive application, such as in ergonomics, sports, rehabilitation, and prosthetic control [1]. A preliminary comprehensive review of various academic databases reveals a remarkable increase in scientific production related to EMG between 2010 and 2023. Specifically, PubMed reported 5444 to 11,382 publications during this period, while Scopus reported 3874 to 8251 articles. Additionally, Google Scholar, leveraging its ability to cover a wider range of sources, identified a significant increase from 1,340,000 publications in 2010 to 3,280,000 in 2023 (search conducted in April, 2024). It is important to note that this basic search strategy returns records of both human and animal studies, which may be relevant depending on the context. This analysis, conducted using a basic search strategy that included only the term EMG, highlights the growing interest and profound commitment of the scientific community in exploring these signals. This phenomenon underscores the increasing relevance of EMG as a subject of study in science and technology.

Unfortunately, the high cost of commercial equipment specialized in biosignal detection, with some models exceeding USD 20,000 [2], poses a significant challenge for research groups with limited resources. This economic barrier limits access to advanced technologies, hindering the development and implementation of innovative solutions across various fields. However, despite these financial constraints, the scientific literature emphasizes several developments where lab-designed prototypes have achieved outstanding results in various applications. These prototypes, though less expensive, have proven effective in areas such as ergonomics, rehabilitation, and prosthetic control, providing accurate and reliable data [3,4,5,6]. These advancements highlight the ingenuity and determination of researchers who continue to push the boundaries of knowledge and technology in EMG and biosignal detection, even in the face of financial constraints.

Within this growing interest, several areas of EMG prototype development stand out; for example, significant progress has been made in creating devices that detect myopathy [7], and prototypes have been developed to implement wireless solutions for data transmission, enabling use in more realistic environments beyond laboratory settings [8]. Additionally, these prototypes have also been employed to record muscle activity and transform the detected signal to its envelope equivalent, which has potential applications in biomechanics [9].

Other authors have described their prototype development for general application [3,4,10,11]. However, the area where these EMG prototypes find the most applications is in controlling prosthetics or robotic arms. In this field, existing research focuses on detecting muscle patterns and how these can be applied to control prototypes with multiple joints, such as robotic prostheses that replicate complex limb movements [12,13,14,15,16]. These studies are notable for their potential to enhance the functionality and autonomy of prosthetic and robotic devices, highlighting the importance of EMG in advancing assistive technologies.

Table 1 summarizes the electrical characteristics considered by various authors in developing EMG circuits, featuring the main component used for amplifying muscle signals, the filters implemented, the final gain, and the data recording resolution through an analog-to-digital converter (ADC). The instrumentation amplifiers (INAs) employed include models such as INA128 [3,5,13], INA118 [6], AD524 [7], INA333 [8], AD621 [9], AD620 [11], and ADS1298 [17], which are crucial for boosting EMG signals due to their high gain and low noise. Various filters were implemented to eliminate unwanted noise and interference, with values ranging from 3 Hz to 21.2 Hz for the high-pass filters and from 324 Hz to 1 kHz for the low-pass filters. Additionally, some prototypes incorporated notch filters, particularly for eliminating power line interference (50/60 Hz). The total system gain varies widely, from 100 to 37,000, depending on the specific requirements of each application. For data recording, different-resolution ADCs are utilized, ranging from 10 bits to 24 bits, affecting the precision of the measurements, and sampling rates reported from 200 to 32 kHz, reflecting the different needs for temporal resolution in studies.

However, to the best of our knowledge, no methodology has been described that employs detailed statistical analyses to compare these prototypes with commercial equipment to validate the reliability of the results obtained. Previous studies have primarily concentrated on visual comparisons or physical synchronizations [18,19]. In these works, the techniques implemented for the extraction of temporal features from EMG signals are particularly noteworthy. Recent studies highlight the significance of techniques used for extracting temporal features from EMG signals, introducing new time domain features like Average Squared Slope (ASS), Average Signal Magnitude (ASM), and Mean Square Root (MSR), alongside conventional features such as Mean Absolute Value (MAV) and Root Mean Square (RMS) [20]. Frequency and time–frequency domain features, including Mean Frequency (MNF), Median Frequency (MDF), and wavelet transform (WT) coefficients, have also been utilized [21,22]. Studies have compared classification algorithms such as k-Nearest Neighbors (k-NNs), Quadratic Discriminant Analysis (QDA), and Linear Discriminant Analysis (LDA) using time and frequency domain features [21]. Additionally, features based on the first difference of the EMG time series have been evaluated and proposed to improve pattern recognition [23,24]. Finally, a tunable Q-factor wavelet transform (TQWT) and an extreme learning machine (ELM) have been applied to detect neuromuscular disorders [25].

This research focuses on the development and validation of a multimodal EMG single-channel, bipolar input prototype board designed in our laboratory, which allows for the stacking of multiple recording boards to increase the number of bipolar channels, up to a maximum of six, enabling multichannel recordings. Feature extraction techniques commonly used were applied to evaluate and compare the performance of our prototype. We propose a standardized methodology for the design and validation of our prototype, aiming to address gaps in the current literature by providing a comprehensive comparison against commercial equipment to ensure the reliability and accuracy of the results obtained.

## 2. Materials and Methods

### 2.1. EMG Hardware Design

The first step in the design process is the simulation and analysis of the proposed circuit, for which National Instruments Multisim 14.0 software was used. This software simulates the behavior of the components under different operating conditions, thus ensuring the accuracy of the design before physical implementation.

To make the simulation as realistic as possible, a feature of this software that allows for loading a text file with two columns of data was used: the first column contains a time vector and the second one contains a scale in voltage levels. This element is Piecewise Linear (PWL), which constructs a waveform with the data from the text file [26,27]. The file contained data recorded during muscle contractions from a previous experiment, which helped calibrate the circuit to match the data recorded by this equipment.

Based on the electrical characteristics reported by other authors, as specified in Table 1, we designed a circuit tailored to our specific needs. This circuit includes the following stages: the first stage of our design incorporates an instrumentation amplifier with a gain of 10, specifically the INA128 developed by Texas Instruments, similar to those used in other studies reported in the literature, as shown in Table 1. This electronic component is available in the simulation software libraries and utilizes Equation (Equation 1), as reported in its datasheet [28], to determine the gain level for amplification.
(1)$$G=1+\frac{50\;k\Omega }{R_{G}}$$

After this stage, two filtering stages were applied. The first one comprised a first-order passive RC high-pass filter, designed to have a cutoff frequency of 20 Hz. However, considering commercial components for the construction of our prototype, it was determined through a Bode diagram that the actual operating frequency was 20.7 Hz. The second filter is a low-pass filter with a proposed cutoff frequency of 300 Hz. However, due to the use of commercial components, the filter exhibited a cutoff frequency of 330 Hz in practical implementation.

Once the signal was filtered, a second amplification stage with a voltage gain of 100 was incorporated. By combining this gain with that of the first stage, a total gain of 1000 was achieved. It was necessary to amplify the EMG signals, which were initially at millivolt levels, to volts, to facilitate their processing by an analog-to-digital converter (ADC). Finally, a fourth-order notch filter at 60 Hz was integrated. For the purposes of this work, we used the ADC integrated into the Biopac® system, recording our conditioned signal as a simple analog input without using the software’s EMG-specific features. These features include predefined frequency ranges and filters optimized for EMG. We chose to bypass them to ensure comparability between the raw signals from our prototype and the Biopac system. Figure 1a represents the sequence of the stages implemented in our prototype, while Figure 1b depicts a front and a back view of the electronic circuit.

To facilitate its use and reduce the number of cable connections needed, we opted for a design that fits the Arduino UNO hardware architecture. This design connects the board with the four characteristic connectors of this ecosystem, enabling our circuit to be powered by the main board and creating a common ground connection between both levels.

The proposed design was carried out using EAGLE (version 9.6.2, Autodesk, Inc., San Rafael, CA, USA) and Fusion (version 2.0.20476, Autodesk, Inc., San Rafael, CA, USA) software, which are both developed by Autodesk. Our design follows the Arduino’s footprint, which allows for its connection with the main boards of this brand or with other compatible boards with the same design. Although the circuit was developed in our laboratory, the manufacturing files (Gerber Files) and component assembly files were managed by a company in China that specializes in manufacturing electronic boards with more than 15 years of experience in the field. This collaboration improved the quality of our prototype, reducing errors in circuit construction and ensuring signal integrity.

### 2.2. Hardware Development for Validation Protocol

The commercial equipment used to compare the EMG data is the Biopac® system (BIOPAC Systems Inc., Goleta, CA, USA), specifically the BIOPAC STUDENT LAB BASIC SYSTEMS MP36 model (BIOPAC Systems Inc., Goleta, CA, USA). This tool supports various laboratory tests and experiments for recording physiological signals, including EMG [29,30,31]. The device can record up to four types of signals from different sources simultaneously and operates with a sampling frequency of 2 kHz, ensuring high-resolution data capture. Several cables are included for data collection, among which are the EMG cables model SS2L and the analog signal cable model SS60L, which were used for data synchronization.

To develop this experiment and compare the data from the Biopac® and our prototype to validate the signals recorded, proper data synchronization was a crucial aspect. Therefore, a system was designed to allow for parallel recording from the same muscle location, ensuring that this information was recorded simultaneously by both devices. To solve this challenge, a second electronic board was developed to obtain reliable data from the same recording location and direct the information to both devices.

The design of the second board ensured a fixed inter-electrode distance of 5 cm, maintaining consistent electrode spacing for precise and reliable muscle signal capture, similar to other EMG recording studies [32,33,34]; Figure 2a illustrates the design of this board, which features female connectors located on the bottom of the board, which attach directly to the electrodes placed over the recording site. At the top, there is an additional pair of male connectors necessary to directly transmit the information to the Biopac® through one of the SS2L cables, where the information was recorded on channel 1. These signals were directly processed by this equipment without interference affecting the nature of the signals.

In our design, it was connected through the electronic board to each of the female connectors. The connector marked as U1 in Figure 2a is a 3.5 mm audio jack. Its function sends the raw signals recorded by the electrodes directly to our EMG prototype. This communication was performed using a 3.5 mm audio cable, compatible with the U1 connector, and its other end was similarly connected to the EMG prototype.

To transmit data from the EMG prototype to the Biopac® device, it was necessary to develop a third electronic board, as shown in Figure 2b, which connects to the EMG board to facilitate its use and reduce connection errors. This board has the same male connectors as the previous board, allowing the Biopac® to record the activity of the muscle signals after the analog processing was performed by the prototype, which were read on channel 2 of the Biopac® through the second SS2L cable. Figure 2c shows a connection diagram of the complete setup, including the position of the board on the forearm where data collection was performed, the EMG prototype, the integration board with the Biopac®, and the connection of all this to a computer.

### 2.3. Volunteers

The experiment involved 18 volunteers, consisting of 9 men and 9 women, with an average age of 20 years and a standard deviation of ±2 years. The participants in this experiment are healthy volunteers with no evident signs of neuromuscular pathology or myopathy. They were selected based on inclusion criteria that ensured the absence of any known medical conditions that could affect myoelectric activity. All participants were students at the Universidad Autónoma de Guadalajara, Mexico, where the research was conducted. Specifically, the experiment was carried out in the university’s Medical Device Development Laboratory. After informing the participants about the study’s activities, they gave their consent by signing an informed consent document, which was approved by the Ethics Committee of the Angel Leaño Hospital in Mexico under registration number CEI/2021/001. All procedures implemented complied with the standards of the Declaration of Helsinki, which sets international ethical standards for biomedical research, ensuring that data privacy and participant rights were rigorously protected.

### 2.4. Data Collection

The muscle signal information was recorded through the second board as in Figure 2c, which functions as an interface between the electrodes and the Biopac®. Simultaneously, the Biopac® recorded the EMG signal acquired by our EMG prototype through the third board, which functions as an interface. This setup allowed us to use the BIOPAC’s ADC to digitize and synchronize the EMG signal obtained by our prototype. In its central part, it features the connector named EMG-OUTPUT, which connects directly with the previous board, measuring the muscle signals after the filtering and amplification. This board distributes the processed signals to three male snap connectors of 3.7 mm, including the ground. This connector is commonly used in manufacturing disposable silver chloride electrodes. These connectors are ideal for the SS60L tip, facilitating the connection between both parts and reducing interference that could affect signal integrity due to any movement.

Finally, to identify the start and end of each instance of muscle activity generated during the task, triggers were generated on the Teensy® 3.2 board. This board features a digital-to-analog converter (DAC), which, depending on the type of movement executed by the interface, sent a different voltage level for a period of 5 ms. These voltage levels did not affect the measurements of the muscle signals from both sources; they were only used as triggers to indicate the start of each trial. The voltage levels ranged from 15 mV to 40 mV, in increments of 5 mV, where 15 mV corresponded to movement 1, 20 mV to movement 2, and so on up to 40 mV, as drawn in Figure 3c. This analog signal was captured by the Biopac® on channel 3 using the SS60L cable, thus allowing all signals to be synchronized over time and enabling independent analysis of the signals.

### 2.5. Experiment Protocol

During the experiment, the participants were asked to remain seated with their backs against the backrest of a chair. In front of them was a table where they could rest their elbow and keep their arm in a 45° position, as shown in Figure 3a. During the experiment, they were required to maintain this position and minimize body movements as much as possible to avoid issues with the equipment measurements. Bipolar surface silver chloride electrodes with a diameter of 2.54 cm (including the adhesive ring) were placed on the forearm to target the muscle group involved in wrist and finger extension. This included the extensor carpi ulnaris, extensor digitorum, extensor carpi radialis brevis, and extensor digiti minimi. Although no specific anatomical landmarks were used, the electrodes were positioned based on visual inspection of the forearm to ensure alignment with the muscle belly. A consistent inter-electrode distance of 5 cm was maintained to capture accurate EMG signals.

In front of the participants was a monitor displaying a visual interface developed with Python (ver 3.2) and the PyGame (ver 2.6.0) libraries. This interface randomly presented six different contraction movements related to hand and wrist motions, including wrist hyperextension, wrist flexion, fist clenching (finger flexion), wrist abduction (radial deviation), wrist adduction (ulnar deviation), and finger extension with abduction, that the participants had to execute. Each movement was repeated 10 times, resulting in the acquisition of 60 trials. Figure 3b demonstrates the different types of movement requested of the participants. Each trial lasted 2 s, with a rest window of 5 s between each one. During each trial, the same Python interface sent commands to a Teensy® 3.2 development board, which sent a different voltage level for a period of 5 ms as a trigger to ensure the correct identification of the simultaneous EMG recordings.

Despite the variety of movements performed by the participants, we do not intend to classify them. Instead, we will focus on extracting a set of features from the EMG signals to directly compare the performance between different EMG recording devices.

**Figure 3 sensors-24-06787-f003:**
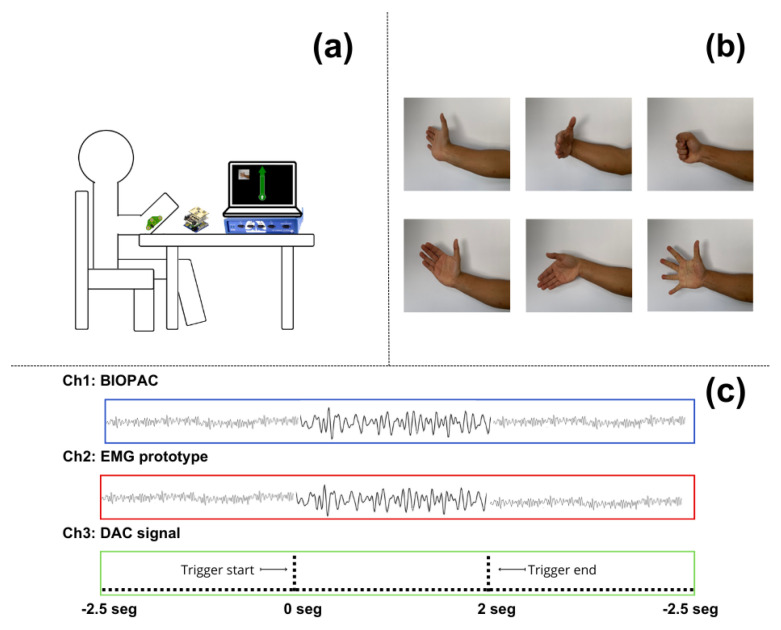
(**a**) Representation of a volunteer during recording, showing the body posture and arm placement during the execution of the trials; (**b**) movements requested by the Python interface that participants had to perform; and (**c**) the time diagram of the recording acquired by the Biopac® device, showing the synchronization of the signals captured by the EMG prototype and the Biopac®, as well as the DAC signal used as a trigger.

### 2.6. Data Processing

The configuration used for data recording with the Biopac® was to set an initial sampling frequency of 2 kHz. It is important to note that its ADC was used to record the signal from our EMG prototype as well, so both signals were sampled at the same frequency. For all data processing stages, including sample rate reduction, filter implementation, and normalization, MATLAB version R2023a (The MathWorks, Inc., Natick, MA, USA) was used. The sampling rate was reduced to 500 Hz. Next, a Butterworth band-pass filter with a frequency range of 20 Hz to 200 Hz [35,36] was implemented to isolate the signal components most relevant to our study. Finally, the data from both devices were normalized on a scale from −1 to 1 using the following equation:(2)$$y=\frac{x}{\max(|x|)}$$
where *y* is the normalized value, *x* is each individual value in the dataset, and max(|x|) is the maximum absolute value in the dataset. This standardizes the signal amplitudes to a common voltage scale, thus facilitating their comparison and subsequent analysis.

### 2.7. Visual Comparison

To analyze and compare the signals recorded by the Biopac® equipment and our EMG prototype, a representative segment of muscle contraction was selected. This segment includes the two seconds before the contraction, the duration of the contraction itself, and up to two more seconds after the contraction.

For a more detailed comparison, the fast Fourier transform (FFT) was applied to both signals after applying the filtering stages. The equation used for the FFT is
(3)$$X\left(k\right)=\sum_{n=0}^{N-1}x\left(n\right)\cdot e^{-j2\pi kn/N},$$
where X(k) is the transform of the signal in the frequency domain, x(n) is the signal in the time domain, and *N* is the total number of samples.

### 2.8. Statistical Analysis

After filtering and normalizing the data from both devices, 60 windows with a duration of 3 s each were extracted for each of the files. These windows correspond to the trials executed by each participant. The selected three seconds include the moment the movement instruction starts until one second after its completion. Although each movement was intended to be executed for two seconds, an additional second was included to account for the delay time participants needed to start and finish the movement.

To compare the performance of our prototype, we used a MATLAB toolbox for extracting features from EMG signals, commonly used in various classification applications [37,38]. This toolbox offers 40 different features. However, for our study, we selected only 22 features that we deemed most relevant for our analysis. The selection was based on several key factors, which are detailed below. Table 2 presents the selected features and their respective equations:Relevance in muscle pattern discrimination: The selected features effectively discriminate different muscle patterns in previous studies. For example, RMS (Root Mean Square) and MAV (Mean Absolute Value) are used as they can capture the energy and intensity of the EMG signal [21,38].Usage in previous studies: Many of the selected features, such as FZC, have been frequently used in previous research for movement classification and muscle contraction detection [37].Ability to capture temporal and spatial information: The selected features include temporal and spatial measures, comprehensively evaluating the EMG signal. For example, WL provides information on the variability and complexity of the signal.Computational efficiency: We considered the computational efficiency of the features, and chose those that offer a good balance between discrimination capability and computational cost. This is crucial for real-time applications such as prosthetic control.

**Table 2 sensors-24-06787-t002:** Names and descriptions of the features used to compare the two EMG signals.

No. Eq.	Name	No. Eq.	Name
1	Zero Crossing Adjustment (FZC)	12	Modified Mean Absolute Value 2 (MMAV2)
2	Enhanced Waveform Length (EWL)	13	Integrated EMG (IEMG)
3	Enhanced Mean Absolute Value (EMAV)	14	Root Mean Square (RMS)
4	Absolute Value of the Sum of Exponential Roots (ASM)	15	Willison Amplitude (WA)
5	Absolute Value of the Sum of Square Roots (ASS)	16	Logarithm Detector (LD)
6	Cardinality (CARD)	17	Mean Absolute Value (MAV)
7	Log Difference of Absolute Standard Deviation (LDASDV)	18	Mean Absolute Deviation (MAD)
8	Log Difference of Mean Absolute Value (LDAMV)	19	Interquartile Range (IQR)
9	Myopulse Rate Percentage (MYOP)	20	Kurtosis (KURT)
10	Order V Measure (VO)	21	Coefficient of Variation (COV)
11	Modified Mean Absolute Value (MMAV)	22	Standard Deviation (SD)

Once the 60 windows for each device were obtained, the 22 functions from the MATLAB toolbox were applied to each of them. To compare both devices, the Mean Absolute Percentage Error (MAPE) was used, a metric that measures the accuracy of a model or device in relation to actual values.
(4)$$\mathrm{MAPE}=\frac{1}{n}\sum_{i=1}^{n}\left|\frac{y_{i}-{\hat{y}}_{i}}{y_{i}}\right|\times 100$$
where $y_{i}$ is the actual value in the *i*-th sample, ${\hat{y}}_{i}$ is the predicted or measured value in the *i*-th sample, and *n* is the total number of samples. This metric is expressed in a range from 0 to 1, where 1 indicates a complete lack of similarity between the values, and 0 points out that both values are the same. To convert this value into a percentage, the result is multiplied by 100.

For practical purposes of our research, we slightly modified the original MAPE equation by subtracting the MAPE value from 1, inverting its logic so that 0 indicates no similarity and 1 indicates high similarity.
(5)$$1\;-\;\mathrm{MAPE}=\left(1-\frac{1}{n}\sum_{i=1}^{n}\left|\frac{y_{i}-{\hat{y}}_{i}}{y_{i}}\right|\right)\times 100$$

## 3. Results

### 3.1. Electrical Simulation and Hardware Development

For the structuring of the design and validation methodology presented in this study, the process included designing and simulating the circuit to define its characteristics based on the parameters of the signal of interest, followed by its fabrication. This was followed by the development of a complete set of hardware and connections designed to facilitate validation. Finally, an experimental task was proposed, and the obtained data were analyzed to validate their similarity.

The results obtained from the simulation of our circuit are shown in Figure 4a. The signal generated by the PWL source with the Biopac® equipment data is signified in red, while the blue signal represents the visual characteristic of the first stage of our circuit, indicating that the signal has been amplified 10 times. The design of the circuit stage that achieves this result is shown.

The Bode plot of the first-order passive filter and the cutoff frequency obtained from the simulation is shown in Figure 4b, and Figure 4c shows the Bode diagram of the low-pass filter, with an exact cutoff frequency of −3 dB and a voltage gain of 0.707 Av. In Figure 4d, the recording of the EMG signal is shown, displaying a relaxation phase from second 0 to second 1, followed by 2 s of sustained contraction, and concluding with another relaxation phase. During the contraction, the muscle signals reached nearly 2 V in amplitude, revealing a significant difference in amplitude levels between this stage and the one shown in Figure 4a. The characteristics of the 60 Hertz notch filter are shown in Figure 4e. At a frequency of 58 Hz, the attenuation is −61 dB or a voltage gain of 0.0021, while at exactly 60 Hz, the attenuation is −34 dB, or a gain of 0.0332. These figures represented a considerable voltage reduction for the frequency generated by power service providers. The design of this filter is pictured in Figure 4e.

The current consumption of the EMG prototype was measured using an open-circuit method with a benchtop multimeter, showing a baseline consumption of 49.05 mA in a resting state and a slight increase to 49.3 mA during muscle contractions. This low variation in energy consumption during operation demonstrates the energy efficiency of the prototype, making it suitable for portable applications and prolonged use with a standard power source, such as a 9 V battery.

Figure 5 illustrates the three electronic circuits that comprise our muscle signal acquisition and synchronization system. Figure 5a presents the EMG prototype, highlighting the various connectors that facilitate interaction between the three boards and the Biopac® equipment. Figure 5b shows the second circuit, which connects directly on top of it and serves as the interface between the Biopac® equipment and the electromyography prototype. The third electronic board, shown in Figure 5c, is positioned directly on the participant’s forearm. This board enables the simultaneous recording of signals from both devices at the same registration point. Figure 5d depicts the integration of the three electronic circuits with the evaluation board selected for this research. Finally, Figure 5e displays a volunteer during the recording protocol, with the entire setup fully implemented. The compact dimensions of the EMG prototype, measuring 53.5 × 66 mm, enhance its portability and ease of use in various settings, ensuring that the device is suitable for both clinical and research environments.

### 3.2. Visual Comparison

To visually represent the comparison between the signals recorded by the Biopac® equipment and our EMG prototype, Figure 6a shows a representative segment of muscle contraction. This segment includes the two seconds before the contraction, the duration of the contraction itself, and up to two more seconds after the contraction. It was observed that both exhibit similar characteristics in response to muscle contraction, although the signal from the EMG prototype shows a higher level of noise.

Furthermore, the change in signal amplitude immediately after executing the digital trigger instruction, indicated by the vertical dotted line in Figure 6a at second 0, was evaluated. The signal amplitude returned to its resting state approximately half a second after two seconds, coinciding with the moment participants were instructed that the contraction had ended.

Moreover, the cross-correlation analysis of the signals yielded a temporal domain correlation of 0.66 with zero lag, indicating that both signals exhibited the highest similarity when they were not time-shifted.

The frequency spectrum, highlighted in Figure 6b, demonstrates that the filters were applied correctly, maintaining the desired frequency range between 20 Hz and 200 Hz. The frequency spectra of both devices show similar amplitudes and frequency ranges.

Additionally, a two-dimensional cross-correlation of the spectrograms of both signals, shown in Figure 6c, was performed, resulting in a correlation of 0.9564 with zero phase shift. This allows us to quantify the quality of the fit between the two signals in terms of explained variability, with the coefficient of determination indicating a 90% explained variance.

### 3.3. Statistical Analysis

The comparison of each selected metric is shown in Figure 7, which presents a heatmap of feature values by subject, illustrating the consistency of the EMG signal analysis across different subjects and functions. Each cell in the heatmap represents the (1 − MAPE) value for a specific function and subject, with darker colors indicating better agreement between the EMG prototype and the Biopac® equipment. A value of 1, or close to it, indicates that the metrics calculated from the signals obtained by our prototype and the commercial device are nearly identical. This result validates that the quality of the recordings obtained from both devices produces an equivalent amount of relevant information.

Figure 8 illustrates the average (1 − MAPE) values for each function used in the EMG signal analysis. The overall average value across all functions is 0.876 (87.6%). Finally, Figure 9 displays the average (1 − MAPE) values for each subject involved in this study. Higher values mean better agreement between the EMG prototype and the Biopac® equipment. The overall average value across all subjects is also 0.876 (87.6%).

## 4. Discussion

In this work, we propose the standardization of a methodology for the design and validation of electrophysiological recording devices. This process encompasses the selection of the board’s features, the simulation of its components using real signals, the manufacturing process to prevent errors during production, and the validation of its performance. The primary focus of this study is the effort to create an experimental environment that allows for the simultaneous recording of signals from both the designed prototype and a commercially accepted device. This approach ensures that the data are obtained from the same registration site, thereby enhancing the comparability of the signals. Typically, adjacent electrodes are used for such recordings, but this method often captures signals from different areas, leading to reduced similarity between the recordings. Additionally, we address the synchronization of both signals by utilizing the ADC of a single device, enabling the temporal synchronization of the recordings. This required the development of an interface between our EMG prototype and the commercial device. Finally, the development of the third board facilitated the synchronization of the recordings with the recording paradigm developed in Python, opting for analog synchronization instead of digital synchronization. This approach allows for the adaptation of this validation protocol to other devices without the need for software integration between paradigms. By using Python, we avoid licensing issues, making the methodology accessible to other research groups.

Additionally, the highest correlation between the signals from both devices was observed without time lag in both the temporal correlation and the spectrogram correlation, confirming that the devices synchronized the recordings correctly. However, the relatively low temporal correlation could be attributed to the sampling system of the commercial device across its various channels. Specifically, if the ADC conversion does not occur at the same temporal instant, the rapid changes in the EMG signal might be recorded at different times relative to the sampling pulse, leading to variations in voltage measurements and thus lowering the correlation coefficient.

Nevertheless, by performing cross-correlation in a time–frequency representation and obtaining a correlation of 0.95, we addressed the temporal complication. This indicates that 90% of the variability in one variable is explained by the variability in the other, allowing us to quantify the “quality” of the fit between two variables in terms of explained variability. Additionally, the absence of a phase shift in the coefficient suggests that there is no time lag between the signals, serving as a second validation method for the synchronization setup designed to validate the EMG, with the 0.95 correlation indicating a high degree of similarity between the signals.

Our EMG prototype achieved an average similarity of 87.6% when compared to an accepted commercially available product (the Biopac). This result is encouraging, as other studies have also achieved good results with low-cost EMG devices. For example, Fuentes del Toro et al. [18] found that their low-cost systems were almost as accurate as commercial ones in measuring muscle activity. Similarly, Bawa and Banitsas [19] designed and validated a low-cost EMG sensor, demonstrating comparable measurements of muscle activity and fatigue to those obtained with commercial equipment.

The high concordance of our prototype indicates that it can be a viable alternative for various applications, such as ergonomics, sports science, and rehabilitation. The possibility of obtaining reliable and accurate EMG signals with an affordable device could facilitate access to this technology for research groups and clinics with limited budgets. Moreover, this would allow for more studies and applications in muscle pattern recognition and prosthetic control. Our Fourier transform analysis exhibits that our prototype and the Biopac® equipment behave similarly in the frequency domain, meaning that our filtering and amplification stages successfully enhance the relevant components of the muscle signal, which is crucial for applications requiring precise signal characteristics.

However, we found that our prototype has higher noise levels than the Biopac® system, which can affect accuracy in some cases. Thus, we plan to reduce this noise in future versions through better shielding and grounding techniques, as well as improved filtering algorithms. Additionally, although this study included 18 volunteers, increasing the sample size and considering participants with different muscle conditions could provide a more comprehensive validation of our prototype. Including a more diverse population would also help to identify potential limitations and ensure that our device performs well in different situations.

Finally, future research should integrate advanced machine learning algorithms into the prototype for real-time muscle pattern recognition. This would enhance its applicability in prosthetic control and rehabilitation settings, providing immediate feedback based on detected muscle activity.

## 5. Conclusions

This work contributes to the advancement of biomedical signal processing technology, offering a reliable and efficient solution through the integration of multiple circuits. Our work highlights the system’s capability to simultaneously record electromyographic signals from multiple devices, and ensures accurate and synchronized data acquisition that is essential for achieving proper validation and comparison with commercial devices and for achieving high concordance. Moreover, this study confirms the effectiveness of our design, while the identified areas for improvement pave the way for future enhancements, which can be further applied in various clinical and research settings. The comprehensive evaluation of the system in real-world scenarios underscores its robustness and potential for broader applications in the field of biomedical engineering.

The development and validation of our EMG prototype represent a significant step toward providing an affordable yet reliable solution for detecting and analyzing muscle signals, contributing to the growing body of literature on low-cost EMG systems and their potential to democratize access to advanced biosignal detection technologies.

## Figures and Tables

**Figure 1 sensors-24-06787-f001:**
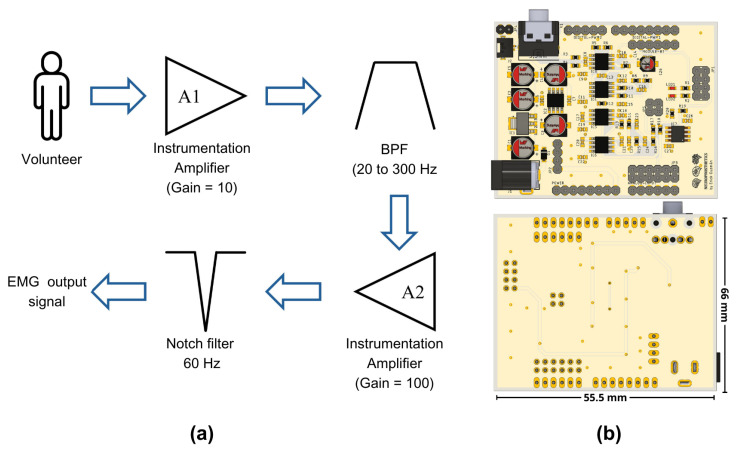
(**a**) Muscle signal processing stages implemented in the EMG prototype and (**b**) the design of the electronic board using EAGLE software; top and bottom views.

**Figure 2 sensors-24-06787-f002:**
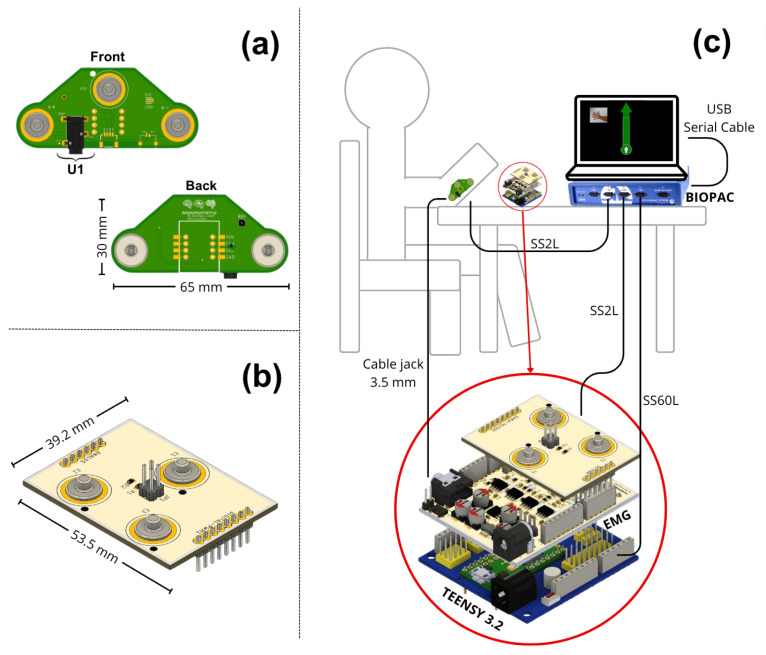
(**a**) Electronic board developed for the connection between the electrodes and the forearm. This board sends raw data to the Biopac® device and the EMG prototype. (**b**) The second electronic board connects to the EMG prototype. It transmits the muscle signal information to the Biopac® through the connectors it has, which are similar to those of the previous board. These connect through the SS60L cable. (**c**) Connection diagram of the experimental elements, showing a representation of a subject with board 1 connected on the forearm, its connection to the Biopac®, and the EMG prototype. It also shows the connection of the prototype to the microcontroller and the second board, its connection to the Biopac®, and the connection between the Biopac® and a computer. The connection between the Teensy 3.2 board and the Biopac® (via the SS60L cable) functions as an analog trigger.

**Figure 4 sensors-24-06787-f004:**
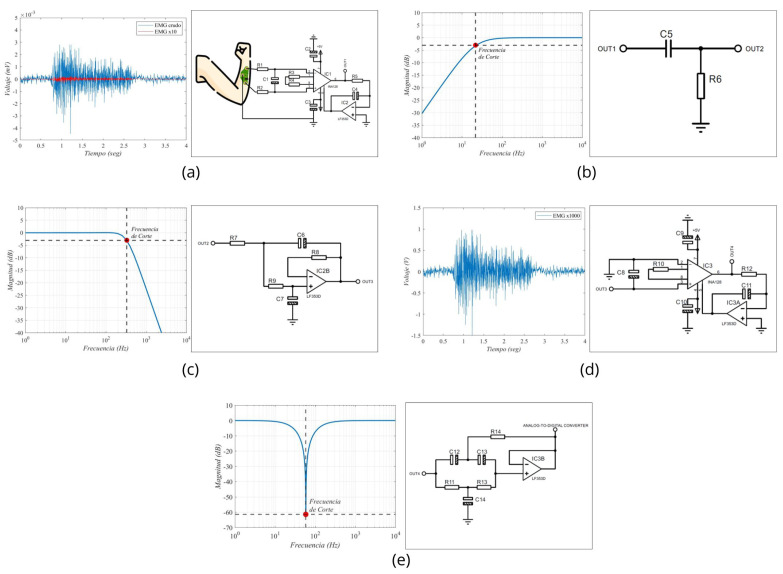
Stages of muscle signal processing implemented in the EMG prototype. (**a**) Diagram of signal pre-amplification using an instrumentation amplifier; (**b**) passive high-pass filter; (**c**) active low-pass filter; (**d**) the second stage of amplification; and (**e**) 60 Hz notch filter.

**Figure 5 sensors-24-06787-f005:**
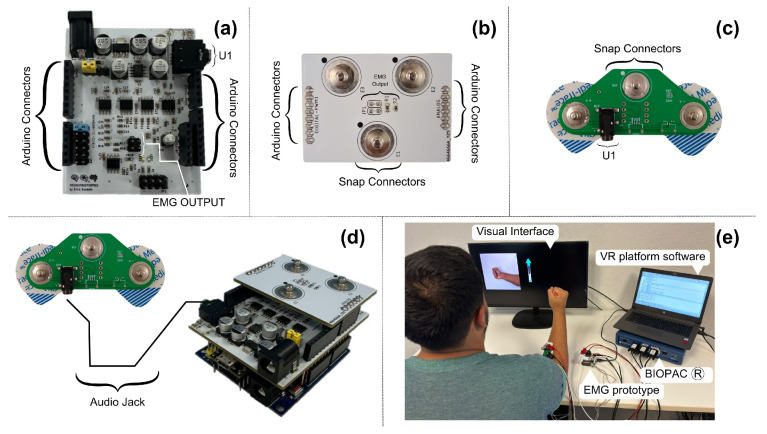
(**a**) The final version of the EMG prototype includes connectors compatible with the Arduino UNO model and connectors for interacting with other electronic boards; (**b**) electronic board for the communication of muscle signals after processing; (**c**) electronic circuit that interacts with the skin surface, indicating the connections of the electrodes and the U1 connector to interact with the EMG prototype; (**d**) connection of all modules; and (**e**) volunteer with the setup during the recording session.

**Figure 6 sensors-24-06787-f006:**
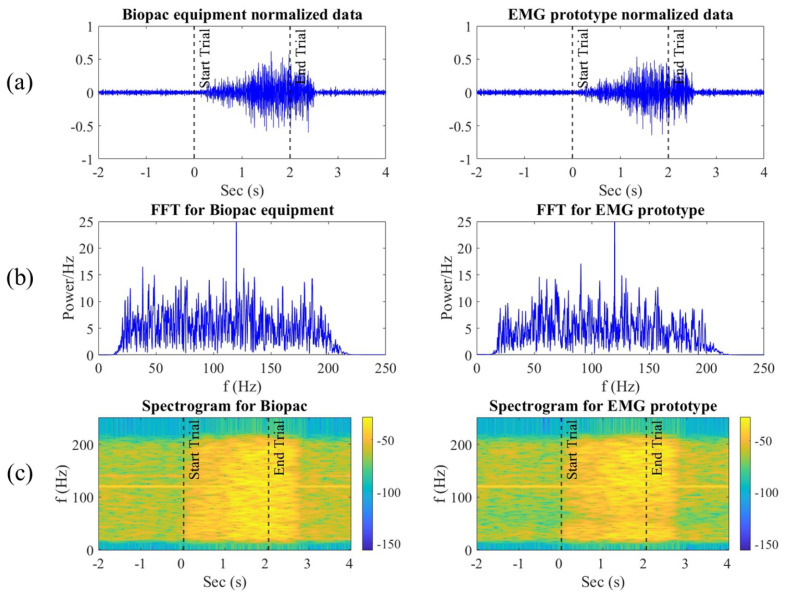
(**a**) EMG signals in the time domain: the left side shows the signal acquired by the Biopac® equipment, and the right side shows the EMG signal recorded with our prototype. The dashed lines indicate the start and end marked with the visual interface. (**b**) Frequency analysis using the Fourier transform after applying a band-pass filter from 20 to 200 Hz. (**c**) Spectrogram of each of the signals.

**Figure 7 sensors-24-06787-f007:**
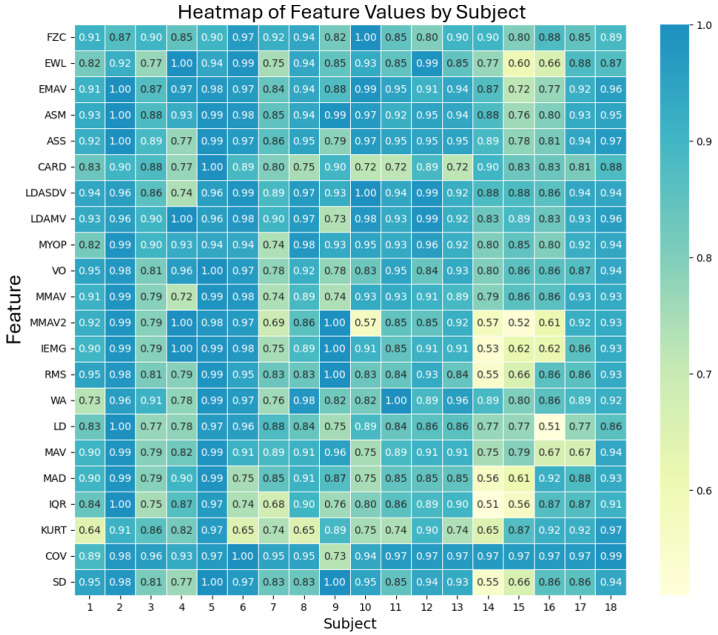
Heatmap of feature values by subject, showing the (1 − MAPE) values for each feature and subject. Darker colors indicate better agreement between the EMG prototype and Biopac® equipment.

**Figure 8 sensors-24-06787-f008:**
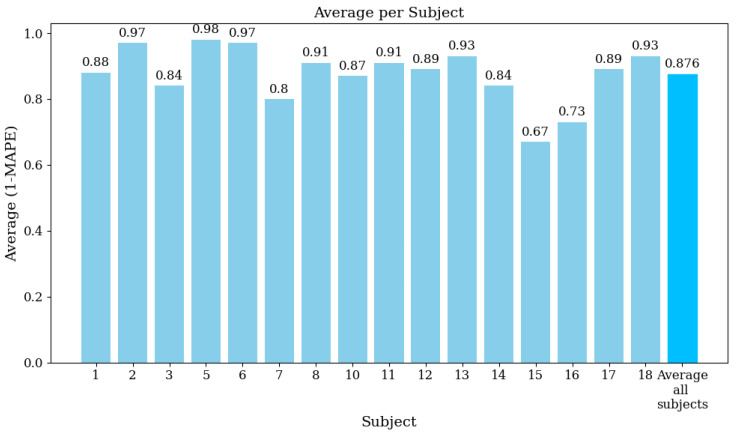
Average (1 − MAPE) per subject, showing the agreement between the EMG prototype and Biopac® equipment for each subject.

**Figure 9 sensors-24-06787-f009:**
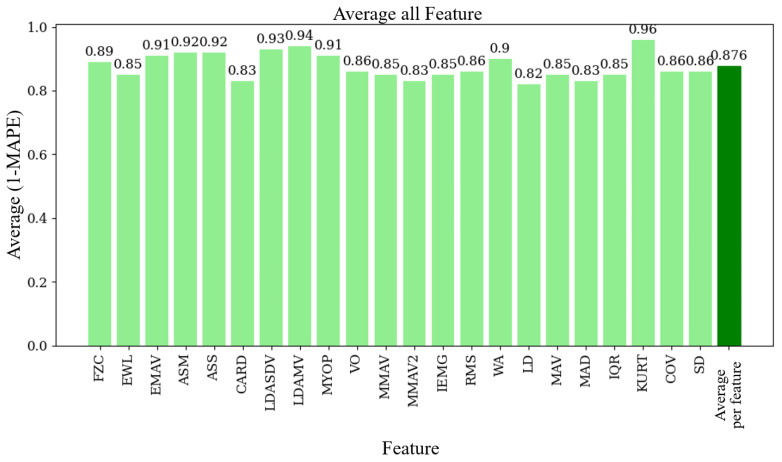
Average (1 − MAPE) per feature, showing the agreement between the EMG prototype and Biopac® equipment for each feature.

**Table 1 sensors-24-06787-t001:** Reported design features used in the development of an EMG circuit.

	Inst. Amplifier	High-Pass Filter	Low-Pass Filter	Gain	Notch Filter	Data Recording	Sample Rate
Fortune et al. [3]	INA128	21.2 Hz	430 Hz	237–3400	-	24 bit ADC	1 kHz
Prakash et al. [5]	INA128	11 Hz	324 Hz	3700	-	NI ELVIS II	2 kHz
Geethanjali et al. [13]	INA128	10 Hz	500 Hz	72–1200	-	Dspace, NI, DATAQ	1000 Hz
Moura et al. [6]	INA118	10 Hz	500 Hz	101	-	10 bit ADC	10 kHz
Barioul et al. [7]	AD524	-	-	1000	-	10 bit ADC	200 Hz
Imtiaz et al. [8]	INA333	20 Hz	450 Hz	100–10,000	50/60 Hz	12 bit ADC	1 kHz
Supuk et al. [9]	AD621	3 Hz	500 Hz	2000	50 Hz	16 bit ADC	1 kHz
Poo et al. [11]	AD620	-	-	2000	50 Hz	24 bit ADC	-
Pancholi et al. [17]	ADS 1298	-	1k Hz	-	-	24 bit ADC	250–32 kHz

## Data Availability

The experimental data can be found at the following GitHub link.

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
