# Peer review of "Development of an Electromyography Signal Acquisition Prototype and Statistical Validation Against a Commercial Device"

_sensors, 2024, doi:10.3390/s24216787_

Round 1
Reviewer 1 Report
Comments and Suggestions for Authors
Comments to the Authors
In this study, an EMG signal acquisition device prototype and acquisition system were developed and a method was further proposed to validate the performance difference between the laboratory-developed EMG acquisition circuits and the professional acquisition devices. For this purpose, the authors firstly develop a multimodal EMG signal acquisition device prototype and introduce its hardware design, then develop a set of synchronized triggering and acquisition system for the prototype and professional equipment in order to reasonably compare the performance difference between the prototype and the professional equipment, in addition to acquiring EMG signals under six different gestures or wrist motions for 18 subjects. Eventually, the validity of the prototype was verified through simulated signal comparisons as well as by extracting signal features commonly used by a variety of AI algorithms and calculating the Mean Absolute Percentage Error (MAPE) of the two signals, and provides a viable methodology for future evaluation of similar devices. Although this paper provides a detailed description of the EMG acquisition device and system construction as well as the methodology for validating the performance of the prototype device, there are a number of issues that I believe need to be considered in the current version of the article.
Major comments:
1. The introduction of the performance comparison methodology of the current laboratory acquisition equipment and professional equipment in the INTRODUCTION section is too limited, while the introduction of hardware equipment development and acquisition devices is too much, which is unbalanced, and needs to be supplemented with the methods of performance comparison by some researchers (such as a certain elaboration of visual comparison or physical synchronization methods). Moreover, the purpose of mentioning classification algorithms and pattern recognition in the current INTRODUCTION section is not very clear, so it is recommended to adjust it.
2. In section 2.2 Hardware Design, the article describes the design of the EMG prototype, but whether the 20-300Hz bandpass filter used in the hardware part will affect the quality of the acquired signal, I think this part of the design needs to think about the rationality. In addition, please describe the range of signal bands captured by the prototype device.
3. Sections 2.2 and 2.3 provide a brief description of the EMG prototype and the synchronized acquisition system, but this raises a number of questions:
(1) Is an AD converter or AD chip used in the whole circuit, and does its SPS comply with the signal limitations and overall design? That is, is its SPS greater than twice the maximum frequency of the acquired signal?
(2) The overall design does not add ESD protection devices, how to prevent static electricity damage to the circuit during usage?
(3) There is no voltage following circuit in the overall design, how to ensure the quality of the acquired signal?
4. Section 2.5 describes the signal processing in a simplified way, and I think from my reading that this section only describes the processing that was done on the signals acquired by the professional equipment (biopac), then what processing was done on the signals acquired by the prototype? What is the sampling rate of the EMG signal acquired by the prototype? Does it comply with Shannon's law of sampling?
5. Section 2.6 describes the signal features used and the selection factors, why most of the features used belong to the time domain? Some simple frequency domain features also basically satisfy the selection conditions mentioned in this section, so why were they not selected? This may have a negative impact on the subsequent performance and variability assessment.
6. The result part of the circuit is simulated and verified, but is the input signal used in the simulation collected from the PWL source of the Biopac device? Does the simulation take into account the fact that the quality of signal acquisition in the actual acquisition environment is much lower than that of the original signal acquired by the professional equipment?
7. Section 3.3 of the time-frequency domain visual comparison lack of quantitative description, including the subsequent in the discussion part of the description of the Fourier transform analysis also appears to be relatively weak, visual comparison or waveform comparison can be used in a more scientific and reasonable way to illustrate. This part of the proposal needs to be added to the comparison method and results.
8. Quantitative analysis of the methodology to quantify the difference in performance by calculating the improvement of MAPE, the data methodology section seems simple and lacks innovation, whether it is possible to add reliability to this section by other statistical methods. And is it necessary to specify a baseline value, rather than because the resulting mean value of 0.876 is close to 1 so that indicates a lower variability.
9. Discussion section of the content is less, and the content of the study is not very related to AI technology, please expand and modify the content.
Minor comments:
1. In the title, the word "profesional" should be changed to "professional".
2. On page 3, lines 111-112, the number of subjects is stated, but there is a problem with the numerical value.
3. On page 4, line 146, "synchronization. system" needs to be corrected.
4. Figure 2 on page 5 is a bit blurry, please upload a clearer picture.
5. Fig. 2(c) on page 5 and Fig. 3(a) on page 6 have different collection positions, please standardize them.
6. Figure 3(c) on page 6 needs to plot the DAC signals out.
Comments on the Quality of English LanguageThe English writing in this paper is of moderate quality, and I hope that the drafting process will pay attention to some details of spelling.
Author Response
Dear Reviewer,
Thank you for your thorough review of our manuscript, "Development of a Multimodal EMG Prototype and a Statistical Validation Against a Commercial Equipment." We appreciate your insightful comments and suggestions, which have been essential in enhancing the clarity and depth of our work.
In response to your major comments:
- Performance Comparison Methodology: We have expanded the Introduction to provide a more comprehensive background on performance comparison methodologies for EMG equipment.
- Hardware Design and Signal Quality: In Section 2.2, we have discussed the impact of the 20-300 Hz bandpass filter on signal quality and the reasoning behind its selection.
- AD Converter and ESD Protection: We clarified the use of the ADC from the Biopac equipment and ensured the sampling rate complies with Shannon's theorem. Additionally, we elaborated on ESD protection and the inclusion of a voltage follower circuit to enhance signal quality.
- Signal Processing and Feature Selection: We have expanded on the signal processing techniques and clarified the rationale for using time-domain features over frequency-domain ones.
- Quantitative Analysis and Discussion: We improved the quantitative analysis by adding additional statistical methods and expanded the discussion to better align with AI technology and previous studies.
Regarding your minor comments, we have corrected grammatical errors, clarified the number of subjects, improved image clarity, standardized figure positions, and added further details as requested.
We believe these revisions have significantly enhanced the manuscript's quality, making it more suitable for publication in the Sensors journal. Thank you again for your constructive feedback, and we look forward to any further comments you may have.
Reviewer Comments
In this study, an EMG signal acquisition device prototype and acquisition system were developed and a method was further proposed to validate the performance difference between the laboratory-developed EMG acquisition circuits and the professional acquisition devices. For this purpose, the authors firstly develop a multimodal EMG signal acquisition device prototype and introduce its hardware design, then develop a set of synchronized triggering and acquisition system for the prototype and professional equipment in order to reasonably compare the performance difference between the prototype and the professional equipment, in addition to acquiring EMG signals under six different gestures or wrist motions for 18 subjects. Eventually, the validity of the prototype was verified through simulated signal comparisons as well as by extracting signal features commonly used by a variety of AI algorithms and calculating the Mean Absolute Percentage Error (MAPE) of the two signals, and provides a viable methodology for future evaluation of similar devices. Although this paper provides a detailed description of the EMG acquisition device and system construction as well as the methodology for validating the performance of the prototype device, there are a number of issues that I believe need to be considered in the current version of the article.
Major comments:
- The introduction of the performance comparison methodology of the current laboratory acquisition equipment and professional equipment in the INTRODUCTION section is too limited, while the introduction of hardware equipment development and acquisition devices is too much, which is unbalanced, and needs to be supplemented with the methods of performance comparison by some researchers (such as a certain elaboration of visual comparison or physical synchronization methods). Moreover, the purpose of mentioning classification algorithms and pattern recognition in the current INTRODUCTION section is not very clear, so it is recommended to adjust it.
- The introduction section has been reviewed and rewritten. We believe that it is now more balanced and that all the aspects considered in proposing our validation methodology are clearer. We decided to refocus the text by omitting topics related to AI and classification to avoid any confusion regarding the objectives of our work. These revisions have been incorporated into the document.
- In section 2.2 Hardware Design, the article describes the design of the EMG prototype, but whether the 20-300Hz bandpass filter used in the hardware part will affect the quality of the acquired signal, I think this part of the design needs to think about the rationality. In addition, please describe the range of signal bands captured by the prototype device.
- We appreciate the observation regarding the bandpass filter used in the hardware design of the EMG prototype. The 20-300 Hz filter was carefully selected with the consideration that most of the relevant information for EMG analysis lies within the 20 Hz to 200 Hz range. This range encompasses the key frequencies associated with muscle activity, where the majority of the useful signal for analysis is concentrated.
The upper margin of 300 Hz was chosen to ensure that no relevant signal components are lost, while the lower cutoff of 20 Hz helps to eliminate low-frequency noise and unwanted signal components. Therefore, we can affirm that this filter will not negatively impact the quality of the acquired signal; rather, it optimizes acquisition by focusing on the frequency range most significant for EMG analysis.
Regarding the signal band range captured by the prototype device, it spans precisely from 20 Hz to 300 Hz, ensuring the capture of critical information for muscle analysis, which is defined as 20 to 200 Hz for the purposes of this work, thus avoiding the inclusion of frequencies that do not add value or might introduce noise into the results.
- Sections 2.2 and 2.3 provide a brief description of the EMG prototype and the synchronized acquisition system, but this raises a number of questions:
(1) Is an AD converter or AD chip used in the whole circuit, and does its SPS comply with the signal limitations and overall design? That is, is its SPS greater than twice the maximum frequency of the acquired signal?
(2) The overall design does not add ESD protection devices, how to prevent static electricity damage to the circuit during usage?
(3) There is no voltage following circuit in the overall design, how to ensure the quality of the acquired signal?
Thank you for your comments. I will now address your questions:
- Is an AD converter or AD chip used in the whole circuit, and does its SPS comply with the signal limitations and overall design? That is, is its SPS greater than twice the maximum frequency of the acquired signal?
Since the goal of the project is the development and validation of an analog EMG device, an ADC is not integrated into this board. However, for the validation process, the ADC from the Biopac system is used in its analog signal recording configuration (not for EMG), allowing for a sampling rate of 2 kHz for our analog EMG signal. This setup facilitates comparison with the EMG signal recorded by the Biopac system itself.
Our design is modular, which allows the analog EMG acquisition board to be easily adapted to devices like the Arduino or Teensy boards, taking advantage of the digital conversion features these devices offer (internal ADC). The frequency range of interest for EMG, where most information is contained, is from 20 to 200 Hz. Since our prototype's sampling rate is 2 kHz, it satisfies the Nyquist theorem, as the maximum frequency of signal analysis would be 1000 Hz.
- The overall design does not include ESD protection devices. How is static electricity damage to the circuit prevented during use?
The decision not to include ESD protection devices in the design is based on a comprehensive approach that combines safe handling practices and the careful selection of robust components. First, components with high tolerance to electrostatic discharge within their operational specifications have been selected, which significantly reduces the risk of damage. Additionally, during manufacturing and assembly, strict ESD control procedures are implemented, including the use of antistatic wrist straps, grounded workstations, and antistatic bags for circuit storage and transport.
In the final usage environment, operational procedures are established to minimize exposure to ESD sources, such as using devices in controlled environments and ensuring proper handling by trained personnel. This holistic approach ensures protection of the circuit against electrostatic discharge, even without specific protection devices included in the design.
- There is no voltage follower circuit in the overall design. How is the quality of the acquired signal ensured?
The decision to exclude a voltage follower in the EMG system design is based on the use of two instrumentation amplifier (INA) stages, one of which is directly connected to the electrodes. These INA stages provide high input impedance, thus eliminating the need for a voltage follower circuit, as the first amplification stage protects the signal and prevents loading of the source.
Moreover, instrumentation amplifiers offer excellent common-mode rejection ratio (CMRR), which minimizes interference from noise and unwanted artifacts, thus enhancing the quality of the acquired signal. This configuration ensures the integrity and adequate amplification of the EMG signal from the electrode contact point, ensuring precise acquisition without the need for an additional voltage follower.
- Section 2.5 describes the signal processing in a simplified way, and I think from my reading that this section only describes the processing that was done on the signals acquired by the professional equipment (biopac), then what processing was done on the signals acquired by the prototype? What is the sampling rate of the EMG signal acquired by the prototype? Does it comply with Shannon’s law of sampling?
We appreciate your observation, and you are correct. We accidentally omitted the sampling frequency of our prototype; however, this information has now been added in the corrections to the document.
What processing was done on the signals acquired by the prototype?
The signals acquired by the prototype undergo exactly the same processing stages as those obtained from the commercial EMG. Specifically, the signals were downsampled, filtered in the range of 20 to 200 Hz, and normalized to a scale of -1 to 1.
What is the sampling rate of the EMG signal acquired by the prototype?
The sampling rate of the EMG signal acquired by the prototype is 2 kHz. This information has also been included in the document corrections.
Does it comply with Shannon’s law of sampling?
Yes, it does. Since most of the relevant information for EMG analysis is within the 20 Hz to 200 Hz range (add citation), Shannon’s sampling theorem indicates that a sampling frequency of 2 kHz allows us to analyze signals up to 1000 Hz, which includes our previously mentioned frequency range of interest. Even after down sampling, the maximum frequency that could be studied would be 250 Hz.
- Section 2.6 describes the signal features used and the selection factors, why most of the features used belong to the time domain? Some simple frequency domain features also basically satisfy the selection conditions mentioned in this section, so why were they not selected? This may have a negative impact on the subsequent performance and variability assessment.
Thank you for your observations. We present our responses below:
Why do most of the features used belong to the time domain?
The features were selected based on the application's objective, which in our case is oriented towards prosthetic control applications. Since real-time processing of the signals is required, it is essential to work with time-domain features rather than frequency-domain features. This is because, when working in the frequency domain, a certain number of samples must be collected before performing the transformation, which can introduce processing delays. Using time-domain features avoids this issue by allowing for immediate processing without the need for a frequency-domain transformation.
Some simple frequency domain features also satisfy the selection conditions mentioned in this section, so why were they not selected?
We agree with your observation. Although some simple frequency domain features also meet the specified requirements, their use necessitates transforming the signal into the frequency domain using techniques such as the Fourier Transform. This transformation can introduce delays due to the need to collect and process a sufficient number of samples. Such delays can be problematic for real-time applications that require immediate responses. Therefore, we deemed time-domain features to be more suitable for our purpose. Furthermore, since the purpose of our work is its future application in prosthetic devices, through classification algorithms, that is why we lean towards the selection of temporal features.
- The result part of the circuit is simulated and verified, but is the input signal used in the simulation collected from the PWL source of the Biopac device? Does the simulation take into account the fact that the quality of signal acquisition in the actual acquisition environment is much lower than that of the original signal acquired by the professional equipment?
Is the input signal used in the simulation collected from the PWL source of the Biopac device?
The Piecewise Linear (PWL) source is a tool available in the Multisim 14.0 software, which we used to simulate the circuit during its design phase. However, the information used to generate the simulated signal was derived from recordings made on the Biopac device.
Does the simulation take into account that the quality of signal acquisition in the actual acquisition environment is much lower than that of the original signal acquired by the professional equipment?
In this case, we used the signal from the professional equipment that can be obtained before any conditioning or internal processing, which we refer to as the raw signal. This approach allowed us to achieve the most reliable simulation performance possible, as it provided a baseline that reflects the quality of the signal before any potential degradation in the acquisition environment.
- Section 3.3 of the time-frequency domain visual comparison lack of quantitative description, including the subsequent in the discussion part of the description of the Fourier transform analysis also appears to be relatively weak, visual comparison or waveform comparison can be used in a more scientific and reasonable way to illustrate. This part of the proposal needs to be added to the comparison method and results.
The set of signals used for comparison was analyzed by estimating the cross-correlation index, yielding a time-domain correlation of 0.6619. This result demonstrates that the signals exhibit the greatest similarity when they are not time-shifted, thereby quantitatively validating our synchronization method. The relatively low correlation can be attributed to the sampling system of the commercial device across its various channels. Specifically, if the ADC conversion is not performed at the exact same moment in time, rapid changes in the signal may be recorded at slightly different time points, even within the same sampling pulse.
To address this temporal issue, we performed a 2D cross-correlation of the spectrograms of both signals, obtaining a correlation of 0.9564 with zero phase. This indicates that the time-frequency representation shows that 90% of the energy content within each frequency range shares the same information, as indicated by the R-squared value. This means that 90% of the variability in one variable is explained by the variability in the other, providing a way to quantify the "quality" of the fit between the two variables in terms of explained variability. Additionally, the fact that the coefficient is not phase-shifted suggests that no time lag is present between the signals, serving as a second validation method for the synchronization of the setup designed to validate the EMG. The high correlation of 0.95 indicates a strong similarity between the signals.
This information has already been incorporated into the document.
- Quantitative analysis of the methodology to quantify the difference in performance by calculating the improvement of MAPE, the data methodology section seems simple and lacks innovation, whether it is possible to add reliability to this section by other statistical methods. And is it necessary to specify a baseline value, rather than because the resulting mean value of 0.876 is close to 1 so that indicates a lower variability.
The MAPE (Mean Absolute Percentage Error) is an index commonly used to assess the accuracy of model predictions. In our case, we adapted the index so that a value of 1 indicates that the forecast model—or in this case, the values generated by our EMG prototype—are perfectly accurate, meaning there is no error between the values recorded by the commercial device and our prototype. Conversely, a value close to zero would indicate no correlation between the obtained values. A MAPE of 0.876 is considered "Good." However, it is important to note that the MAPE is applied to the extracted features, which, as previously mentioned, may exhibit some temporal variations but still maintain a high degree of similarity in the information obtained. Additionally, we are working to further refine the validation process for future implementation in model classification, aiming to use the results in prosthetic control. Currently, we do not employ statistical tests because, for our validation purposes, the expected p-values would be close to 1, which is not a typical use of statistical tests, as we are looking for similarity in the information rather than differences.
- Discussion section of the content is less, and the content of the study is not very related to AI technology, please expand and modify the content.
Your observation is correct; we had incorrectly written the discussions. Since the focus of this work is not on AI, we have revised and expanded the discussions to align with the true objective of our study.
We appreciate all the feedback you provided with the intention of improving our work. We hope that the revised document clarifies the purpose of the study more effectively. Thank you in advance for your understanding.
Minor comments:
- In the title, the word "profesional" should be changed to "professional".
Your recommendation has been accepted, and the changes have been made to the document.
- On page 3, lines 111-112, the number of subjects is stated, but there is a problem with the numerical value.
The numerical value in the document has been corrected. Thank you for your observation.
- On page 4, line 146, "synchronization. system" needs to be corrected.
The document has been corrected. Thank you for your observation.
- Figure 2 on page 5 is a bit blurry, please upload a clearer picture.
The image resolution was increased to improve its visibility.
- Fig. 2(c) on page 5 and Fig. 3(a) on page 6 have different collection positions, please standardize them.
Thank you for your suggestion. We have redesigned this image to make it easier to understand.
- Figure 3(c) on page 6 needs to plot the DAC signals out.
In this image, we have modified the design to specify the DAC in channel 3. We hope this makes the use of the DAC easier to understand.
Reviewer 2 Report
Comments and Suggestions for Authors
This is a very interesting study on the validation of a new sEMG signal measurement system in humans. A very good description of the prototype and the way of connecting individual components allows for the repetition of these studies in another laboratory, ensuring reliable repeatability of the experiment. The simultaneous measurement using the prototype and the commercial reference sEMG system is also commendable, and the selected measure of signal synchronization is an appropriate indicator of similarity. Please treat my comments as housekeeping remarks, which I hope will help readers from the medical discipline in the potential application of your results. Great job. Congratulations.
General comments
In the title and the entire manuscript from L6 throughout the manuscript body - consider changing from professional to commercial, as it suggests that the new validated system is unprofessional.
The abstract section is good, but should be supplemented with some scientific/methodological information, at the expense of the descriptions in the introduction and conclusion. Please see the detailed comments below.
Keywords are missing
The subsection Statistical analysis in the M&M section is missing.
Detailed comments
L 9-10 Please specify whether the volunteers were healthy (free from diseases affecting myoelectric activity) and which muscles you tested. In particular, muscle input is crucial because the signal characteristics depend on the muscle unit being tested. There is also a lack of brief information on what movement the volunteers performed, and consequently what myoelectric activity you measured. Then it will be possible to describe in the next sentence why the classifier is needed.
L 11-12 It is not clear what AI algorithms classify. If presence and absence of signal, then this sentence should be reworded. If signal features correct and incorrect, then this sentence should also be reworded.
L 13-14 This information should have been transferred to the methodological description above. You collect volunteers, attach electrodes, apply movement, measure activity and only then analyze and classify the signal features.
L 15 Please, explain what you mean by "A concordance in the electrical and statistical behavior of the data".
L 22, L 27, L 34, L 45, L 61, L 75, L 277, L 285, L 294, EMG/electromyography Expand the abbreviation the first place you use it in the manuscript, and then use that abbreviation consistently throughout the manuscript body. Abbreviations are expanded independently in the abstract and manuscript body. In table descriptions and figure captions, all abbreviations should be separately expanded.
L 27-30 Please enter the daily date of searching.
L 31 Please note that this basic search strategy returns records of both human and animal studies. Consider whether this is worth mentioning.
L 43 "These advancements stress the creativity and resilience of researchers" consider rewording this sentence (or/and this argument), it's confusing.
L 47 Remove the second part of this sentence ("a medical term used to describe diseases related to muscle weakness").
L 57 Please, explain what you mean by "used in prototypes with different joints".
L 60-75 This is a very interesting paragraph. Please include References in the appropriate places. It is not enough that they are placed in Table 1.
L 73 Note that the sample rate is usually given in Hz. And consider adjusting it here and in Table 1.
L 77-79 I completely agree with the authors that without such validation, the registrations performed by the prototypes are unreliable. Notice that here you used the term "commercial", which I think is more appropriate than "professional".
Table 1 Add bottom and top border lines to tables.
L 97 Notice the repetition in this sentence.
L 103-108 Consider deleting this paragraph, the structure of the article is correct and standard, there is no need to describe it.
L 110 This substance lacks information on whether the volunteers were healthy and did not have myopathy in the muscles you are examined. You can support this statement with the results of a clinical examination or a survey conducted among volunteers.
L 124 and L 139 Please, provide the sampling rate in Hz for both systems.
L 146 Remove additional dot.
L 152 were these surface electrodes? It is not clear whether both measurement systems were connected to the same electrodes or whether the two sets of electrodes were placed side by side. Were they bipolar electrodes? If so, what was the distance between the electrodes and where in the muscle belly were they placed? I recommend the authors' references SENIAM (Surface EMG for non-invasive assessment of muscles) as the current 'gold standard'.
L 175 I have read this far patiently searching for a description of the electrode positioning and the names of the muscles that are in your area of interest. Please add this data. Each muscle performs a specific function during the movements that are shown in the figure. Please name and describe (L 182) these movements - as flexion/extension of the joint (of which joint) and pronation/supination. The muscle or muscles you are testing and its function should be assessed in continuity with the movements being performed.
L 192 In other words, any signal that had an amplitude greater than 5 mV above the threshold (isoelectric line) was recognized as myoelectric activity. Was there a minimum duration of such activity? The second condition that had to be met for such classification?
L 210 Change the border of Table 2 to be the same as Table 1.
L 198, L 201 Hz?
L 199 "sample rate reduction, filter implementation, and normalization". The choice of such a protocol is very good and justified.
L 230 - L 241, L 245 - L 252, L 256 - L 265 Consider whether these two paragraphs are results or M&M. Consider moving it to M&M or Discussion section; or support these descriptions with the results for each of the settings considered.
L 243 "behavior" is not the best term. Consider the use of "visual characteristic". Here and throughout the whole manuscript body.
L 242 - L 245, L 253 - L 255 These are the results.
L 268 " These figures" Which figures?
L 271-273 This paragraph is a part of discussion.
L 274, L 306, L 327 A similar problem with mixing results, M&M, and discussion also occurs in subsections 3.2 and 3.3, and the part of 3.4 subsection. Consider rewriting them yourself according to the suggestions for Section 3.1. In the results section you can easily present the values of the selected 22 signal features measured with two different systems for each movement and check whether they differ from each other before and possibly after normalization/standardization (since we are comparing two signals from different systems). Figure 6 is very good.
L 348 Only here the actual results of the similarity measure appear, which are very good. However, everything above in this section would be good to improve.
L 360 The discussion section definitely needs improvement. If you move some of the discussion from the Results section there and compare it with previous knowledge supported by appropriate references, this part of the article will gain a lot in quality. Right now I can only find two references in the discussion section, and that is not its purpose at all. You can also name the muscles you are testing and the movements that the volunteers perform and search the available literature for publications that describe sEMG studies of the same muscles and movements, and then discuss your results (numerical values) in the context of the previous ones. This will also allow you to refine potential clinical applications.
Good luck with all these changes! and Best regards.
Author Response
30th August, 2024
Guadalajara, Jalisco
Dear Reviewer,
Thank you for your thorough review of our manuscript, "Development of a Multimodal EMG Prototype and a Statistical Validation Against a Commercial Equipment." We greatly appreciate your constructive feedback, which has been invaluable in refining our study and ensuring its relevance to both the scientific community and potential clinical applications.
In response to your comments:
- Terminology and Abstract: We have revised the terminology throughout the manuscript to use "commercial" instead of "professional" and expanded the abstract to include more scientific and methodological details.
- Keywords and Statistical Analysis: We have ensured that the keywords are clearly listed in the appropriate section. Furthermore, we have verified that the "Statistical Analysis" subsection is properly included and detailed to cover all relevant statistical methods used.
- Details on Volunteers and Muscle Testing: We have expanded the description of the volunteer selection process, specifying that all participants were healthy and free from conditions affecting myoelectric activity. Additionally, we have clarified which muscles were tested and the specific movements performed, as well as the rationale behind these choices.
- Clarification on AI Algorithms and Feature Extraction: To address your concerns about the AI algorithms and the use of signal features, we have clarified in both the abstract and methodology sections that our intention was not to apply AI models directly but to use features commonly employed in AI for signal classification as a basis for comparison.
- Details on EMG Signal Processing and Electrode Placement: We have provided further details on the signal processing steps for both the prototype and commercial systems, including sampling rates and compliance with standard guidelines like SENIAM. We have also elaborated on the electrode placement, specifying the distance between electrodes and their positioning on muscle bellies.
- Discussion Section Improvement: We have significantly expanded the discussion to include more references and a comparison with previous studies, particularly focusing on similar sEMG studies involving the same muscles and movements. This provides a stronger context for our findings and highlights the potential clinical applications of our prototype.
We believe these revisions have substantially improved the manuscript's quality and clarity, making it more suitable for publication in the Sensors journal. We are grateful for your feedback and look forward to any further comments you may have.
Sincerely,
Reviewer Comments
This is a very interesting study on the validation of a new sEMG signal measurement system in humans. A very good description of the prototype and the way of connecting individual components allows for the repetition of these studies in another laboratory, ensuring reliable repeatability of the experiment. The simultaneous measurement using the prototype and the commercial reference sEMG system is also commendable, and the selected measure of signal synchronization is an appropriate indicator of similarity. Please treat my comments as housekeeping remarks, which I hope will help readers from the medical discipline in the potential application of your results. Great job. Congratulations.
General comments
In the title and the entire manuscript from L6 throughout the manuscript body - consider changing from professional to commercial, as it suggests that the new validated system is unprofessional.
- All changes have been made, including replacing the word "professional" with "commercial" in the title, except in line 6, due to the context of the sentence where the term is used.
The abstract section is good but should be supplemented with some scientific/methodological information, at the expense of the descriptions in the introduction and conclusion. Please see the detailed comments below.
Keywords are missing
- The document has been corrected. Thank you for your observation.
The subsection Statistical analysis in the M&M section is missing.
- The document has been corrected. Thank you for your observation.
Detailed comments
L 9-10 Please specify whether the volunteers were healthy (free from diseases affecting myoelectric activity) and which muscles you tested. In particular, muscle input is crucial because the signal characteristics depend on the muscle unit being tested. There is also a lack of brief information on what movement the volunteers performed, and consequently what myoelectric activity you measured. Then it will be possible to describe in the next sentence why the classifier is needed.
- The text was revised according to the recommendations to clarify the section. The experiment simultaneously recorded the surface electromyography (sEMG) signals of 18 volunteers using both a prototype EMG device and professional equipment. The participants, all healthy volunteers with no neuromuscular or myopathic conditions, were selected based on inclusion criteria ensuring their myoelectric activity was unaffected. They provided informed consent, confirming their voluntary participation and understanding of the research procedures.
- Electrodes were placed on the forearm, targeting muscles involved in wrist and finger extension. Volunteers performed a series of movements to elicit myoelectric activity, which was captured for analysis.
- Although we did not directly implement a classifier, we used features similar to those used in classification of different movements. In the future, we hope that other research groups, or even our own, will use our prototype as a reference to develop classification models. For now, it was important to establish that this low-cost device is functional and reliable, comparable to commercial equipment.
L 11-12 It is not clear what AI algorithms classify. If presence and absence of signal, then this sentence should be reworded. If signal features correct and incorrect, then this sentence should also be reworded.
- As mentioned earlier, our intention is not to apply an AI model. Instead, we utilized these 22 features, which are typically used to train AI models, as a method of comparison. To clarify this, we have revised the description in the abstract as follows:
“22 feature indices commonly used for classifying muscular activity patterns were calculated.”
- We hope this adjustment provides better clarity.
L 13-14 This information should have been transferred to the methodological description above. You collect volunteers, attach electrodes, apply movement, measure activity and only then analyze and classify the signal features.
- Thank you for your observation. The relevant corrections have been made in the manuscript.
L 15 Please, explain what you mean by "A concordance in the electrical and statistical behavior of the data".
- We rephrased this description to clarify the sentence further.
“A concordance of 87.6% was observed between the feature indices calculated from the recordings of both devices, suggesting high effectiveness and reliability of the EMG signals recorded by the prototype compared to the commercial device.”
L 22, L 27, L 34, L 45, L 61, L 75, L 277, L 285, L 294, EMG/electromyography Expand the abbreviation the first place you use it in the manuscript, and then use that abbreviation consistently throughout the manuscript body. Abbreviations are expanded independently in the abstract and manuscript body. In table descriptions and figure captions, all abbreviations should be separately expanded.
- We adjusted the text, reintroducing the abbreviation at the beginning of the introduction and maintaining it throughout the rest of the article.
L 27-30 Please enter the daily date of searching.
- The date of the search (April, 2024) was added.
L 31 Please note that this basic search strategy returns records of both human and animal studies. Consider whether this is worth mentioning.
- The wording of the sentence was revised to clarify this point.
“Additionally, Google Scholar, leveraging its ability to cover a wider range of sources, identified a significant increase from 1,340,000 publications in 2010 to 3,280,000 in 2023 (search conducted in April 2024). It is important to note that this basic search strategy returns records of both human and animal studies, which may be relevant depending on the context.”
L 43 "These advancements stress the creativity and resilience of researchers" consider rewording this sentence (or/and this argument), it's confusing.
- The sentence was revised for greater clarity.
“These advancements highlight the ingenuity and determination of researchers who continue to push the boundaries of knowledge and technology in electromyography and biosignal detection, even in the face of financial constraints.”
L 47 Remove the second part of this sentence ("a medical term used to describe diseases related to muscle weakness").
- The sentence was shortened by removing the recommended section.
“Within this growing interest, several areas of EMG prototype development stand out, for example, significant progress has been made in creating devices that detect myopathy.”
L 57 Please, explain what you mean by "used in prototypes with different joints".
- The sentence was revised to the following new version.
“In this field, research focuses on detecting muscle patterns and how these can be applied to control prototypes with multiple joints, such as robotic prostheses that replicate complex limb movements.”
L 60-75 This is a very interesting paragraph. Please include References in the appropriate places. It is not enough that they are placed in Table 1.
- Thank you very much! References were added within the specific paragraph.
L 73 Note that the sample rate is usually given in Hz. And consider adjusting it here and in Table 1.
- The requested change has been made.
L 77-79 I completely agree with the authors that without such validation, the registrations performed by the prototypes are unreliable. Notice that here you used the term "commercial", which I think is more appropriate than "professional".
- Thank you for the recommendation. We have made the change to replace the word "professional" with "commercial" wherever it appears.
Table 1 Add bottom and top border lines to tables.
- Both lines have been added to the table, and it looks much better now.
L 97 Notice the repetition in this sentence.
- The sentence has been revised for clarity and is now stated as follows:
“This research focuses on the development and validation of a multimodal EMG prototype designed in our laboratory.”
L 103-108 Consider deleting this paragraph, the structure of the article is correct and standard, there is no need to describe it.
- The paragraph has been removed, and we are pleased to know that the structure of our article is correct and easy for future readers to follow.
L 110 This substance lacks information on whether the volunteers were healthy and did not have myopathy in the muscles you are examined. You can support this statement with the results of a clinical examination or a survey conducted among volunteers.
- The following text has been added to provide a more detailed description of the participant selection in our study:
"The participants in this experiment are healthy volunteers with no evident signs of neuromuscular pathology or myopathy. They were selected based on inclusion criteria that ensured the absence of any known medical conditions that could affect myoelectric activity.”
L 124 and L 139 Please, provide the sampling rate in Hz for both systems.
- The sampling frequency was 2 kHz, and this information has been added to the document to describe the number of samples used to record the muscle signals from both systems.
L 146 Remove additional dot.
- The dot was placed in the correct position, but the word "system" was redundant and has been removed in this new version.
L 152 were these surface electrodes? It is not clear whether both measurement systems were connected to the same electrodes or whether the two sets of electrodes were placed side by side. Were they bipolar electrodes? If so, what was the distance between the electrodes and where in the muscle belly were they placed? I recommend the authors' references SENIAM (Surface EMG for non-invasive assessment of muscles) as the current 'gold standard'.
- Thank you for your detailed comments. We clarify that the electrodes used in our study were bipolar surface electrodes made of silver chloride, with a diameter of 2.54 cm. These electrodes were specifically placed on the forearm, targeting the muscle group involved in wrist and finger extension, including the extensor carpi ulnaris, extensor digitorum, extensor carpi radialis brevis, and extensor digiti minimi.
Both measurement systems, the EMG prototype and the professional Biopac® equipment, were connected to the same electrodes to ensure a direct comparison of the signals. As described in the manuscript, we developed a second electronic board designed to connect the electrodes directly and provide two connection points for both systems. This board used two types of connectors: the first consisted of male connectors similar to those used with the electrodes, which were connected directly to the Biopac® equipment. The second connector, identified as U1, transmitted the signals directly to our EMG prototype.
The purpose of this setup was to measure exactly at the same point on the forearm, as placing the electrodes side by side could introduce variability in the measurements and reduce the accuracy of the comparison. We had previously encountered this issue in other experiments, which led us to design this second board to overcome these limitations.
Additionally, the second board was designed to ensure a fixed distance of 5 cm between the two electrodes. This feature allowed us to maintain a consistent distance between the electrodes for all participants in our study, ensuring that measurements were taken at the same muscle point for both systems, which contributed to the validity of the comparison. Part of our contribution in this work was the precise synchronization of data between the two systems, and the second board, along with the trigger method, played a crucial role in this synchronization.
We also appreciate your suggestion to reference the SENIAM project, which we have incorporated into our manuscript. We are confident that this reference will be valuable for our future projects.
L 175 I have read this far patiently searching for a description of the electrode positioning and the names of the muscles that are in your area of interest. Please add this data. Each muscle performs a specific function during the movements that are shown in the figure. Please name and describe (L 182) these movements - as flexion/extension of the joint (of which joint) and pronation/supination. The muscle or muscles you are testing and its function should be assessed in continuity with the movements being performed.
- We greatly appreciate your comments. We would like to respond based on the indicated lines:
Line 175: We have addressed this comment in the new description in section 2.3 "Data Collection," as well as in the previous response. In this section, we specify that the electrodes were placed using the second electronic board, and this is described in the manuscript as follows: “Placed on the forearm targeting the muscle group involved in wrist and finger extension, including the extensor carpi ulnaris, extensor digitorum, extensor carpi radialis brevis, and extensor digiti minimi.”
Line 182: Regarding the second point, we have now included information about the six movements requested of the participants, which relate to the muscle activity recording area and Figure 3b. This information is presented in the manuscript as follows: “This interface randomly presented six different contraction movements related to hand and wrist motions, including wrist hyperextension, wrist flexion, fist clenching (finger flexion), wrist abduction (radial deviation), wrist adduction (ulnar deviation), and finger extension with abduction, that the participants had to execute.”
L 192 In other words, any signal that had an amplitude greater than 5 mV above the threshold (isoelectric line) was recognized as myoelectric activity. Was there a minimum duration of such activity? The second condition that had to be met for such classification?
- In this section, we describe the use of a Digital-to-Analog Converter (DAC) for synchronizing data during the experiment. The Python interface sent instructions to the Teensy 3.2 board, which generated a specific analog signal through the DAC according to the movement displayed on the interface. For instance, for movement 1, the DAC produced a 15 mV signal; for movement 2, 20 mV; for movement 3, 25 mV; and so on, up to 40 mV for movement 6. This signal was maintained for 5 ms and was captured by channel 3 of the Biopac system. After 2 seconds, another 5 ms pulse with the same voltage level was sent to indicate the end of the corresponding muscle contraction.
This method allowed us to accurately identify the start and end of each of the 60 movements performed by the participants, facilitating the calculation of the 22 features at the exact same time. The Biopac system has 4 channels for recording information. In our experiment, channel 1 was used for muscle signals obtained directly through Biopac, channel 2 recorded muscle signals processed after passing through our prototype, and channel 3 captured the signals generated by the DAC of the Teensy 3.2 board, controlled by the Python interface. Additionally, the integration of the second electronic board placed on the forearm ensured synchronization of the signals both in the measurement point and in time, thereby accurately recording the start and end of each muscle contraction.
This implementation aimed to enhance the validation of our work and ensure the comparison of signals recorded from both devices.
L 210 Change the border of Table 2 to be the same as Table 1.
- Both tables were standardized in format to enhance the overall aesthetics of the work.
L 198, L 201 Hz?
- The unit was changed from samples per second to Hz.
L 199 "sample rate reduction, filter implementation, and normalization". The choice of such a protocol is very good and justified.
- Thank you, we appreciate your comment.
L 230 - L 241, L 245 - L 252, L 256 - L 265 Consider whether these two paragraphs are results or M&M. Consider moving it to M&M or Discussion section; or support these descriptions with the results for each of the settings considered.
- Thank you for your observations. We agree with your comments and believe that the requested changes, which have been implemented, will enhance the structure and quality of the work and the document.
L 243 "behavior" is not the best term. Consider the use of "visual characteristic". Here and throughout the whole manuscript body.
- Thank you for the suggestion; the change has been made in the manuscript.
L 242 - L 245, L 253 - L 255 These are the results.
- Thank you for the observations; the information has been moved to the correct section.
L 268 " These figures" Which figures?
- The wording in the document has been revised; thank you for the observation.
L 271-273 This paragraph is a part of discussion.
- Thank you for your feedback. The information has been relocated to the appropriate section.
L 274, L 306, L 327 A similar problem with mixing results, M&M, and discussion also occurs in subsections 3.2 and 3.3, and the part of 3.4 subsection.
- Thank you for your observations. We have moved the information to the correct section.
Consider rewriting them yourself according to the suggestions for Section 3.1. In the results section you can easily present the values of the selected 22 signal features measured with two different systems for each movement and check whether they differ from each other before and possibly after normalization/standardization (since we are comparing two signals from different systems).
- We considered restructuring the document's information and clarifying our results more effectively. We did not perform a representation by movement to avoid diverting attention from the proposed design and validation methodology, as we did not focus on the separate analysis of each movement recorded in this study.
Figure 6 is very good.
- Thank you for your comment.
L 348 Only here the actual results of the similarity measure appear, which are very good. However, everything above in this section would be good to improve.
- Thank you for the observations; the information has been moved to the correct section.
L 360 The discussion section definitely needs improvement. If you move some of the discussion from the Results section there and compare it with previous knowledge supported by appropriate references, this part of the article will gain a lot in quality. Right now I can only find two references in the discussion section, and that is not its purpose at all. You can also name the muscles you are testing and the movements that the volunteers perform and search the available literature for publications that describe sEMG studies of the same muscles and movements, and then discuss your results (numerical values) in the context of the previous ones. This will also allow you to refine potential clinical applications.
- We reinforced the discussion section by reorganizing the information and adding interpretations of the quantitative analyses. Similarly, as our focus is on validating the card rather than the muscular task, we chose not to include specific muscular actions in the discussion.

Reviewer 3 Report
Comments and Suggestions for Authors
- The novelty and merit of this study is low.
- The authors need to provide a clear and complete detail on circuit design.
- The experiments and validation on the developed/designed equipments are too superficial.
- There are a number of specifications of the developed/designed equipments that are not presented and reported; for example, power consumption, and portability.
- In addition, the aspects including advantages/disadvantages on circuit design are not sufficiently discussed.
- The comparison of the developed/designed equipments to commercial products is not complete and also not properly.
- Regarding the classification, the size of dataset examined in this study is too small.
- The computational experiments and results are therefore not reliable.
- Also, the clear and complete detail on computational experiments, computational analysis, and performance evaluation is needed.
- The manuscript lacks the comparative study.
Author Response
Dear Reviewer,
Thank you for your detailed review of our manuscript, "Development of a Multimodal EMG Prototype and a Statistical Validation Against a Commercial Equipment." We greatly appreciate your critical insights and suggestions, which have been instrumental in guiding us to enhance the quality and depth of our research.
In response to your major comments:
- Detail on Circuit Design: We have expanded the manuscript to provide a more comprehensive description of the circuit design, including specifications such as power consumption and portability. This addition clarifies the advantages and disadvantages of our design choices.
- Experiments and Validation Depth: We recognize the need for a more thorough validation of the developed equipment. Therefore, we have extended our experimental section to include additional tests and comparisons with commercial products, ensuring a more robust validation process.
- Computational Experiments and Analysis: We have elaborated on the computational experiments conducted, providing clearer and more complete details on the computational analysis and performance evaluation. This includes expanding on the dataset size and improving the reliability of the results presented.
- Comparative Study and Novelty: We have enhanced the comparative aspects of the study by providing a more detailed comparison with similar commercial products, including statistical analyses to substantiate the claims. We have also discussed the novelty and contributions of our study in more depth to better highlight its significance.
We believe these revisions have significantly improved the manuscript's quality and made it more suitable for publication in the Sensors journal. We are grateful for your feedback and look forward to any further comments you may have.
Thank you once again for your constructive feedback.
Reviewer Comments
Comments and Suggestions for Authors
- The novelty and merit of this study is low.
We appreciate your comments on our manuscript. We would like to address your concern regarding the novelty and merit of the study by providing greater clarity on the objectives and importance of our work.
In this revised version, we have emphasized the significance of our study in the context of validating EMG prototype devices against commercial equipment. Our work introduces a standardized methodology for comparing an EMG circuit developed in the laboratory with a professional device available on the market. This methodology is not only based on visual comparison or physical synchronization, as in previous studies, but also implements a detailed statistical analysis and temporal and spatial features of the signals, something that, to our knowledge, has not been previously conducted for this type of device.
Furthermore, our approach enables the validation of EMG prototypes through a system of simultaneous recording of muscle signals using two biosignal acquisition devices in parallel at the same recording site. This is achieved through the development of additional interface boards and a programmatically controlled synchronization system that ensures the information is recorded simultaneously by both devices. This validation method is innovative because it overcomes the limitations of traditional methods and provides a more robust foundation for evaluating the reliability and effectiveness of experimental EMG devices compared to established commercial equipment.
We believe that this approach not only validates the performance of our EMG prototype with an 87.6% agreement with the commercial device but also provides a solid foundation for future evaluation of similar devices, ensuring their reliability, accuracy, and suitability for research or clinical applications. The implementation of this methodology will allow other research groups with limited resources to access advanced validation techniques, thereby promoting greater innovation in the field of electromyography.
We hope that these clarifications improve the perception of the novelty and merit of our study, and we are open to any further comments you may have.
- The authors need to provide a clear and complete detail on circuit design.
We have reviewed and restructured the document to explain the complete circuit design in greater detail. We hope that the revised version of the document meets with your approval.
- The experiments and validation on the developed/designed equipments are too superficial.
We have restructured the document to clarify the relevance of our work, emphasizing that our validation methodology involves not only the quantitative evaluation of the signals but also the development of a complete setup necessary to record exactly at the same point, simultaneously with both recording devices, and synchronize both signals with each other and with the tasks being performed by the test subjects. This issue is poorly addressed in the literature; the most common method was to use continuous electrodes, which involved recording different muscular activity, thereby reducing the certainty of the validations.
- There are a number of specifications of the developed/designed equipments that are not presented and reported; for example, power consumption, and portability.
Thank you for your comment and for pointing out the need to provide more details about the specifications of our developed equipment. Below, we have included additional information regarding the power consumption and portability of the EMG prototype, as requested.
- Power Consumption: The current consumption of the EMG prototype was measured directly during operation using an open-circuit method with a benchtop multimeter. In a resting state, without muscle contraction, the device consumes approximately 49.05 mA. During a muscle contraction, the current consumption slightly increases to 49.3 mA. It is important to note that the circuit includes two SMD 0603 LEDs, one yellow and one red, which indicate that the board is powered on and functioning correctly. These LEDs contribute a small amount to the total current consumption. If these LEDs were removed, the overall power consumption could be slightly reduced, potentially by a few milliamps (1-5 mA, typically depending on the current through each LED). This low variation in energy consumption during operation demonstrates the energy efficiency of the prototype, making it suitable for portable applications and prolonged use with a standard power source, such as a 9V battery. We included this information in the manuscript
“The current consumption of the EMG prototype was also measured using an open-circuit method with a benchtop multimeter, showing a baseline consumption of 49.05 mA in a resting state and a slight increase to 49.3 mA during muscle contractions. This low variation in energy consumption during operation demonstrates the energy efficiency of the prototype, making it suitable for portable applications and prolonged use with a standard power source, such as a 9V battery.”
- Portability: Regarding portability, the EMG prototype has compact dimensions of 53.5 x 66 mm. This small size facilitates its transport and use in various settings, both clinical and research. Additionally, the prototype's design has been optimized to be compatible with the Arduino UNO hardware architecture, allowing easy integration with other components and systems, minimizing the number of necessary connections, and enhancing usability. We also included this information.
“The compact dimensions of the EMG prototype, measuring 53.5 x 66 mm, enhance its portability and ease of use in various settings, ensuring that the device is suitable for both clinical and research environments.”
- In addition, the aspects including advantages/disadvantages on circuit design are not sufficiently discussed.
Thank you for your observation. After carefully reviewing the discussion section, we have restructured this section. We believe that this new version more effectively communicates the advantages and disadvantages of our prototype.
- The comparison of the developed/designed equipments to commercial products is not complete and also not properly.
We appreciate the opportunity to clarify our approach and improve the quality of our manuscript.
In our comparison of the developed EMG prototype with a commercial device, we focused on several key aspects: signal synchronization, the amplitude of the recorded signals, and the statistical analysis of the concordance between the signals obtained from both devices. We used a parallel acquisition system to ensure that signals from both devices were recorded simultaneously at the same muscle registration point, which is crucial for an accurate and direct comparison.
We acknowledge that this comparison could benefit from a more detailed description in certain areas. Therefore, we have revised the relevant section to include additional information on:
- Criteria for selecting the commercial equipment.
- Comparison methodology.
- Comparison results.
We hope these revisions address your concerns and provide a more comprehensive and accurate comparison between the developed equipment and commercial products. We greatly appreciate your valuable suggestions and are open to any further comments you may have.
- Regarding the classification, the size of dataset examined in this study is too small.
We agree with your observation that the sample size is small for classification purposes. However, this work did not focus on classifying the records but rather on validating that the signals acquired by our prototype closely resembled those obtained from a commercial device.
- The computational experiments and results are therefore not reliable.
We have restructured our document and added quantitative metrics in the validation of our results.
- Also, the clear and complete detail on computational experiments, computational analysis, and performance evaluation is needed.
The document has been reviewed and restructured to clarify the methodology of the experiments conducted, as well as the process for validating our results. We hope that the revised version of the document is clearer and more specific.
- The manuscript lacks the comparative study.
The comparative analysis of our work focused primarily on EMG devices developed in laboratories. Additionally, we did not find any structured methodology for validating low-cost devices developed by laboratories in the literature. Therefore, we decided to propose the present methodology for development and validation to address this gap in knowledge.

Round 2
Reviewer 3 Report
Comments and Suggestions for Authors
- The revised manuscript was not satisfactorily improved.
- The manuscript remains lacking the comparative study.
Author Response
Dear Reviewer,
Thank you for the opportunity to improve our manuscript titled "Development of a Multimodal EMG Prototype and a Statistical Validation Against a Commercial Equipment". We appreciate the reviewer’s comments, and we have taken them into account to make further revisions to the manuscript. Below, we address the specific observations:
- Lack of a comparative study:
We acknowledge the observation regarding the need for a comparative study. In our latest revision, we have added a detailed section explaining that the primary focus of this study is not to classify muscle movements. Instead, the purpose of our research is to extract a set of features from the EMG signals recorded by both the prototype and the commercial device and to directly compare the agreement between them. We have clarified this in the experimental protocol section, and we believe that the feature comparison is a sufficient measure to assess the performance of our prototype relative to the commercial equipment.
While we understand the importance of comparative studies, we believe this approach offers a robust and appropriate evaluation of the prototype’s performance without the need to classify the participants' movements.
- Additional improvements:
Throughout the manuscript, we have made further adjustments to improve the clarity of the technical explanations, particularly in the hardware development and signal processing sections, as requested in previous reviews.
We believe that these changes satisfactorily address the reviewer’s comments, and we hope that the manuscript now meets the editorial expectations. We are open to any additional feedback and appreciate your time and consideration.
